# COVID-19 Mortality in Public Hospitals in a Brazilian State: An Analysis of the Three Waves of the Pandemic

**DOI:** 10.3390/ijerph192114077

**Published:** 2022-10-28

**Authors:** Larissa Soares Dell’Antonio, Franciéle Marabotti Costa Leite, Cristiano Soares da Silva Dell’Antonio, Camila Brandão de Souza, Juliana Rodrigues Tovar Garbin, Ana Paula Brioschi dos Santos, Nésio Fernandes de Medeiros Junior, Luís Carlos Lopes-Júnior

**Affiliations:** 1Secretaria de Estado da Saúde do Espírito Santo, Special Epidemiological Surveillance Nucleus, Instituto Capixaba de Ensino, Pesquisa e Inovação (ICEPi), Vitória 29010-120, ES, Brazil; 2Graduate Program in Public Health, Federal University of Espírito Santo (U.F.E.S.), Vitoria 29047-105, ES, Brazil; 3Hospital Sírio-Libanês, Instituto de Ensino e Pesquisa, São Paulo 01308-060, SP, Brazil; 4Hospital Universitário Cassiano Antônio de Moraes, Vitória 29041-295, ES, Brazil

**Keywords:** coronavirus infections, COVID-19, mortality, public health surveillance, epidemiologic monitoring, health management, public hospitals

## Abstract

Objective: To analyze COVID-19 deaths in public hospitals in a Brazilian state, stratified by the three waves of the pandemic, and to test their association with socio-clinical variables. Methods: Observational analytical study, where 5436 deaths by COVID-19 occurred in hospitals of the public network of Espírito Santo, between 1 April 2020, and 31 August 2021, stratified by the three waves of the pandemic, were analyzed. For the bivariate analyses, the Pearson’s chi-square, Fisher’s Exact or Friedman’s tests were performed depending on the Gaussian or non-Gaussian distribution of the data. For the relationship between time from diagnosis to death in each wave, quantile regression was used, and multinomial regression for multiple analyses. Results: The mean time between diagnosis and death was 18.5 days in the first wave, 20.5 days in the second wave, and 21.4 days in the third wave. In the first wave, deaths in public hospitals were associated with the following variables: immunodeficiency, obesity, neoplasia, and origin. In the second wave, deaths were associated with education, O_2_ saturation < 95%, chronic neurological disease, and origin. In the third wave, deaths were associated with race/color, education, difficulty breathing, nasal or conjunctival congestion, irritability or confusion, adynamia or weakness, chronic cardiovascular disease, neoplasms, and diabetes mellitus. Origin was associated with the outcome in the three waves of the pandemic, in the same way that education was in the second and third waves (*p* < 0.05). Conclusion: The time interval between diagnosis and death can be impacted by several factors, such as: plasticity of the health system, improved clinical management of patients, and the start of vaccination at the end of January 2021, which covered the age group with the higher incidence of deaths. The deaths occurring in public hospitals were associated with socio-clinical characteristics.

## 1. Introduction

Based on the impact that the rapid spread of a disease, COVID-19 (Coronavirus Disease 2019), caused by a virus could have on underdeveloped countries with poor health infrastructure, the World Health Organization (WHO) on 30 January 2020, declared a health emergency of international concern. Later, the WHO recognized it on 11 March 2020, as the COVID-19 (Coronavirus Disease) pandemic whose causative pathogen was the new coronavirus SARS-CoV-2 [1].

COVID-19 was first identified on 1 December 2019, in Wuhan City, the capital of Hubei Province in China, when it was observed that a group of people had symptoms of a pneumonia of unknown cause. The one thing in common was the fact that they went regularly to a seafood market in the city itself. The cases quickly spread throughout the country, besides expanding to other countries, and become a worldwide concern [2].

By mid-April 2020, more than two million cases and 120,000 deaths had already been reported worldwide; although in Brazil, by that time, approximately 21,000 confirmed cases and 1200 deaths had been reported [3].

The first Brazilian case was identified on 26 February 2020, in the state of São Paulo and the first death on 17 March of the same year, also in the same state [4]. In the state of Espírito Santo (ES), the first case was recorded on 26 February, the Ash Wednesday holiday, and its disclosure was made on 5 March by the Ministry of Health [5].

Among the Brazilian regions, the South stands out for its high incidence (13,421.3 cases/100,000 inhabitants), followed by the Midwest (13,330.9 cases/100,000 inhabitants), North (9897.8 cases/100,000 inhabitants), Southeast (9075.5 cases/100,000 inhabitants), and Northeast (8299.6 cases/100,000 inhabitants). As for mortality, the Midwest region ranks first (338.0 deaths/100,000 inhabitants), followed by the Southeast (309.5 deaths/100,000 inhabitants), South (301.6 deaths/100,000 inhabitants), North (249.6 deaths/100,000 inhabitants), and Northeast (202.2 deaths/100,000 inhabitants) [6].

A survival analysis study developed in the state of Rio Grande do Norte showed higher risks of death by COVID-19 in individuals aged 80 years or older (HR = 8.06; *p* < 0.001), male (HR = 1.45; *p* < 0.001), with non-white skin color (HR = 1.13; *p* < 0.033) or without information (HR = 1.29; *p* < 0.001), who had comorbidities (HR = 10.44; *p* < 0.001) or did not inform of the presence of comorbidities (HR = 10.87; *p* < 0.001) [7].

Similarly, a study conducted in the state of ES with the objective of analyzing the factors associated with death in hospitalized individuals observed higher mortality in the elderly, patients with comorbidities, and users of public hospitals [8].

The first study conducted in the state of Espírito Santo, Brazil which aimed to analyze the survival of patients hospitalized by COVID-19 with the source of data from the state’s official Health Surveillance Information System, have shown that the risk factors for death from COVID-19, such as the age group 60 to 79 years, and 80 years or more, non-work infection, patients with comorbidities, especially cardiovascular, chronic kidney disease, chronic neurological disease, smoking, obesity and neoplasms were associated with lower survival [9]. A cross-sectional study undertaken in Kazakhstan, identified 80% of cases of COVID-19 infection were asymptomatic or mild, and among the symptoms, the most common were cough (20.8%) and sore throat (17.1%), fever (11.6%) and runny nose (7.2%). In addition, the presence of cardiovascular diseases, such as hypertension, diabetes, obesity, respiratory, and maternal diseases have affected the mortality rate significantly in the country [10].

The number of cases and deaths reported for COVID-19 is directly dependent on the testing policy adopted by the location [11,12]. Some countries perform testing only in the most severe cases requiring hospitalization. Others recommend testing everyone who has symptoms. There are still those who implement mass testing. Particularly in Brazil, the confrontation of the pandemic had to cope with the political scenario and the lack of integrated national planning policies within the states and municipalities [12,13].

Given this unfavorable scenario, the use of existing information in information systems can be a good tool for monitoring the epidemic, planning prevention, and controlling measures, as well as for evaluating the impact of this new virus on the morbidity and mortality profile in Brazil [13].

In this sense, the present study aimed to analyze COVID-19 deaths in public hospitals in a Brazilian state, stratified by the three waves of the pandemic, and its association with socio-clinical variables.

## 2. Materials and Methods

This is an observational, analytical study in which the population were patients living in the state of Espírito Santo, Brazil, affected by COVID-19 and who died in public hospitals between 1 April 2020 and 31 August 2021. Data were collected from the internal control database of hospitalizations and deaths in the state, monitored and fed by the technical team of the COVID-19 Health Situation Room, and provided by the State Department of Health of Espírito Santo, Brazil.

The dependent variables of the study were COVID-19 deaths in the three waves of occurrence, recorded in the period between 1 April 2020 and 4 September 2020 (first wave), 5 September 2020 and 13 February 2021 (second wave), and 14 February 2021 and 31 August 2021 (third wave). The occurrence of the wave of COVID-19 mortality was defined as the beginning of an upward mortality trend with a significant evolution, reaching its plateau and tending to fall, with its end marked by the lowest recorded incidence of death until the beginning of a next upward movement.

We considered as independent variables: socio-clinical characteristics among which: socioeconomic and demographic (age group, gender, race/color, education, origin), exposure factors and risk conditions (ICU admission, chronic lung disease, chronic cardiovascular disease, chronic kidney disease, chronic liver disease, diabetes mellitus, immunodeficiency, HIV infection, neoplasia (solid or hematological tumor), smoking, bariatric surgery, obesity, tuberculosis, neoplasms, chronic neurological disease), signs and symptoms (fever, difficulty breathing, nose wing flapping, intercostal screening, cyanosis, O_2_ saturation < 95%, coma, cough, sputum production, nasal or conjunctival congestion, coryza, sore throat, difficulty swallowing, diarrhea, nausea or vomiting, cephalea, irritability or confusion, adynamia or weakness, pharyngeal exudate, conjunctivitis, seizure, loss of smell, loss of taste), confirmation criteria, and the time between diagnosis and death.

The program STATA version 15.1 (StataCorp^®^, College Station, TX, USA) was used to organize the data. For statistical analysis, the program IBM SPSS Statistics (SPSS^®^ Inc., Chicago, IL, USA) version 24 was used.

The Kolmogorov–Smirnov test was used to assess the probability distribution and normality of the data. Pearson’s chi-square test verified the relationship between the socio-clinical variables and the incidence of the death waves. When this test did not meet its requirements (n > 20, all expected values in the table are greater than 1 and at least 80% of these are greater than or equal to 5), we used Fisher’s exact test. The Friedman test compared the time between diagnosis and death between the waves, and the simple quantile regression related the time between diagnosis and death in each wave of death incidence. The advantages of this regression include: (i) it is required when the distribution is not Gaussian; (ii) it is robust to outliers; (iii) when residuals are not normal and/or non-homoskedastic, they produce more efficient estimators than those of the ordinary least squares regression and it is more informative, not only being restricted to an average, since the regression can be obtained by the median [14].

The Hosmer–Lemeshow test was used to evaluate the fitted model comparing the observed and expected frequencies and the Omnibus Test was used to assess whether the explained variance in the dataset is significantly greater than the unexplained variance.

A multiple multinomial regression with the forward variable selection method associated the public link with socio-clinical variables. The alpha level of significance used in all analyses was 5%.

The study was approved by the Research Ethics Committee under opinion no. 4,166,025 on 21 July 2020.

## 3. Results

Of the 1,810,128 notifications for COVID-19, a total of 566,385 (31.28%) were confirmed for COVID-19 in which 11,720 evolved to COVID-19 death, of which 5436 occurred in public hospitals.

The mean time between diagnosis and death in public hospitals was 18.5 days in the first wave, 20.5 days in the second wave, and 21.4 days in the third wave. The test rejected the null hypothesis of normal distribution for the time between diagnosis and death; therefore, the nonparametric technique is the most adequate (Kolmogorov–Smirnov; significant if *p* ≤ 0.05).

Table 1 shows the deaths that occurred in public hospitals in accordance with the three waves. Among the deaths, one can notice a higher occurrence of elderly people, brown (parda) race, elementary school education, difficulty breathing, without cyanosis, with cough, without sputum production, without nasal or conjunctival congestion, and without coryza. There was also a higher occurrence among those who died of cephalea, irritability or confusion, neoplasia, obesity, neurological disease, and were not smokers. The predominant diagnostic confirmation was laboratorial. The higher number of deaths was found in people from the metropolitan region (*p* < 0.05).

The variables fever, O_2_ saturation < 95%, and chronic cardiovascular disease showed different behaviors in the three waves. In the first and third wave, among the deaths we observed a higher occurrence of symptoms such as fever, which did not happen in the second wave. On the other hand, O_2_ saturation < 95% and chronic cardiovascular disease were more prevalent in the first and second wave of COVID-19 deaths.

In order to evaluate the association of the public bond with socio-clinical variables for the three waves of death incidence, a comparison was made between the data from public hospitals and the grouped data from the other hospital bonds (private and philanthropic).

Evaluating the first wave of COVID-19 mortality in public hospitals, it can be seen that among the deaths, there is a 1.8 times greater chance of having immunodeficiency compared to those who do not present with this condition. In addition, among the cases of deaths, an increased chance of 1.26 and 4.0 times of not having obesity and neoplasms was found, respectively (Table 2). In the second wave (Table 3), a higher chance of illiterate patients (OR = 3.53) compared to those with higher education is observed in public hospitals among the deaths. In addition, exposure to oxygen saturation lower than 95% also showed higher odds of COVID-19 mortality (OR: 1.54). Another finding is that among the deaths, there is a 1.68 times higher chance of people who did not have chronic neurological disease. In the third wave (Table 4), among deaths in public hospital, it was found that there is approximately 2.0 times greater chance of deaths of people of black and/or yellow race/color compared to those of white race/color. In addition, there is a higher chance of death of illiterate (OR = 2.3) individuals and those with elementary education (OR = 2.6), compared to those with higher education (Table 4).

Table 4 also presents the clinical profile in the third wave. It can be seen that among the deaths in public hospital, patients exposed to difficulty breathing (OR: 1.49) and diabetes mellitus (OR: 1.25) have a greater chance of dying. Another point to highlight is that, among the deaths, the greatest chance of dying was observed in people who did not have nasal or conjunctival congestion (OR = 1.50), irritability or confusion (OR = 2.58), adynamia or weakness (OR = 1.23), chronic cardiovascular disease (OR = 1.34), and neoplasms (OR = 3.53).

It is worth noting that in the first wave of COVID-19 deaths, the greatest chances of death were for people coming from the Central and North Region (OR: 5.4), while in the second and third waves, this group presented odds of 3.9 and 3.6, respectively, compared to those coming from the South Region (Table 3).

## 4. Discussion

To the best of our knowledge, this is the first study conducted in Brazil that analyzes COVID-19 mortality in public hospitals, besides bringing stratified information between the three different waves of the pandemic.

Among the main findings, in the first wave, statistical significance was observed for the following variables: immunodeficiency, obesity, neoplasms, and place of origin. In the second wave, an association was observed with education, oxygen saturation lower than 95%, and chronic neurological disease. In the third wave, the association was verified with race/color, education, difficulty breathing, diabetes mellitus, nasal or conjunctival congestion, irritability or confusion, adynamia or weakness, chronic cardiovascular disease, and neoplasms. It is worth noting that the greater chances of death in public hospital are among people coming from the Central and North Region, in all waves, in relation to those from the South Region.

When analyzing the different waves, we observed an increase in the time between diagnosis and death; however, statistical significance was observed only for the third wave in which the median interval was 20 days. Several interrelated factors may be associated with the increased interval between diagnosis and death, such as: the plasticity of the health system, which at this time was already more structured to receive these patients; improved clinical management of patients due to greater knowledge and safety of health professionals; and also, the start of vaccination at the end of January 2021, covering the age range in which a higher incidence of deaths had been observed. It is worth noting that the state government had restricted the operation of various activities during this period (quarantine) in order to contain transmission and, thus, reduce the number of hospitalizations, preventing the collapse of the hospital network and providing better care to patients.

It is noteworthy among the results that regardless of the wave of incidence of the COVID-19 pandemic, the elderly undoubtedly represented the group with the greatest impact. This finding is in line with numerous studies showing the most prevalent death outcome in older people [15,16,17,18,19,20].

In Brazil, the epidemiological situation of COVID-19 mortality has also been shown to be related to demographic and income aspects, that is, social factors [21]. In this context, we understand such a relationship as a result of the impact of the variable race/color upon the calculation, which is a complex one, because it is not limited to biological or genetic factors, but, above all, it represents a set of meanings and sociocultural exposures that portray inequity in health [22]. The incidence of deaths occurring differently among the different races is pointed out in several studies as a product of the existing social inequality and the difficulty of access to health services by the black and brown population [23,24].

In Brazil, information on color/race is sometimes missing as well, as it is not provided at the time of registration since there is a tendency towards whitening in the country [25,26], which hinders interpretations in research and population surveys and, consequently, the reformulation of public health policies. Considering COVID-19, it is observed that in public hospitals, yellow, black, and brown people had the highest chances of death compared to whites. In general, the rates of disease and death by COVID-19 for the black population have been two to three times higher than for whites. This is due, among other causes, to the difference in access to public or private hospitals with adequate structure, public hospitals being more used by the black population [23]. Racism is a social determinant of health, impacting the health of populations and representing the root cause of inequities in access to goods, resources, and opportunities [27].

Another important variable in the health equity process is education. Like the variable race/color, information on patients’ education is sometimes neglected at the time of registration [28], which causes gaps and interpretation biases. According to Andrade da Silva (2021) [29], the COVID-19 spread was higher among the poor and less educated. In the present study, we observe a higher chance in public hospitals of deaths among those with lower education.

A study [30] points out a wide variation in hospital mortality by COVID-19 in SUS, associated with demographic and clinical factors, social inequality, and differences in the structure of services and the performance of health services. There is consensus that the highest prevalence of hospitalization due to COVID-19 occurs among people with lower levels of school education, with the highest proportion of deaths corresponding to those with no education [31]. According to the COVID-19 and Sustainable Development Report, released by the United Nations (UN), skin color and education are key determinants of the lethality rates of the disease among Brazilians [32].

With regard to signs and symptoms, difficulty breathing and O_2_ saturation < 95% were two variables statistically associated with the outcome under study. The Report of the WHO–China Joint Mission 2019 [33] pointed out that patients with severe disease had, among other symptoms, difficulty breathing/dyspnea [20,34], respiratory rate ≥ 30/min, and blood O_2_ saturation ≤ 93%. In Northern Italy, patients with the worst prognosis had a higher median respiratory rate and dyspnea more frequently since their admission, which was the main factor associated with a severe prognosis [20].

In this context, it is noteworthy that difficulty breathing in the third wave of the pandemic was a very present symptom among patients according to the fourth edition of the Guiding Manual for Combatting the Pandemic (Guia Orientador para o Enfrentamento da Pandemia) in the Health Care Network organized by the National Council of Health Secretaries [35]. This guide also discusses reservations about patients with acute chronic conditions, pointing out that this comorbidity is an important risk factor for COVID-19 mortality; its contents provide actions and activities that Primary Health Care needs to develop in order to control this comorbidity in their patients [35].

Cough, sore throat, sputum production, nasal or conjunctival congestion, and coryza are considerably common symptoms in COVID-19, and although some were related to the outcome in the bivariate analysis, they were not retained in the multivariate analysis, except for nasal congestion (OR: 1.50). However, it is important to note that these symptoms are important, as shown in a study conducted in Kazakhstan, which found that although 80% of cases of COVID-19 infections were asymptomatic or mild, among the most common symptoms were cough (20.8%) and sore throat (17.1%), fever (11.6%), and coryza (7.2%) [10]. In China, the most common symptoms on admission of cases were fever and cough, followed by sputum production and fatigue [15].

The present study identifies, among the deaths in public hospitals, a higher chance in patients who did not present with irritability or confusion, as well as neurological disease. However, it is important to consider the importance of identifying signs of delirium, as it may serve as a relevant marker for identifying patients with COVID-19 at risk for poor outcomes, including ICU admission and death, which has been difficult to manage because some methods of treating COVID-19 are inherently deliriogenic [36]. Further, the incidence of mental alteration or stroke on admission predicts a modest but significantly higher risk of in-hospital mortality regardless of the severity of illness [37,38].

A noteworthy finding in the present research was the higher chance of exposure to diabetes mellitus (OR: 1.25) among the deaths occurring in public hospitals in Espírito Santo. A literature review by Iranian researchers found that the presence of comorbidities, such as cardiovascular disease, obesity, diabetes, hypertension, and pulmonary disease, acted as accelerators for the progression to a worse prognosis [39]. In addition, in a hospital in Wuhan, China, comorbidities were present in almost half of the patients, and hypertension was the most common comorbidity, followed by diabetes and coronary heart disease [15].

The presence of cardiovascular disease, hypertension, diabetes, obesity, respiratory and kidney disease affected the mortality rate significantly in Kazakhstan, with cardiovascular disease increasing the probability of the outcome death by 4 times, diabetes increased it by 2.4 times, kidney disease by 5.9 times and respiratory disease by 2.6 times [10]. In the same vein, a study conducted with about 89,000 hospitalizations that occurred from February to June 2020, showed a greater chance of death among patients with comorbidities [30].

Although the study did not show a higher chance of death among those exposed to neoplasia, it is important to note that cancer patients are more susceptible to infections in general, due to the systemic immunosuppressive state caused by treatments, and are likely to have an increased risk of a worse prognosis. In China, patients with various types of cancer, particularly hematological and lung neoplasms, were more likely to develop severe complications of COVID-19 [40] with an advanced disease stage being an even greater aggravating factor [41]. However, in line with our results, a study conducted by a group of researchers from the National School of Public Health at Fiocruz also found higher mortality among immunosuppressed people [30].

There was no evidence of greater chances of death among those exposed to obesity; however, researchers in the School of Public Health at the University of Washington highlighted the disproportionate impact of COVID-19 in obese and severely obese patients, considering the relationship of obesity versus lung function. Decreased expiratory reserve volume, functional capacity, and compliance of the respiratory system are associated with obesity, and in increased abdominal obesity lung function is even more impacted, considering the decreased diaphragmatic excursion when in the supine position. It is also possible to associate the overexpression of inflammatory cytokines, which occur in a state of obesity, which contributes to increased morbidity when COVID-19 infection occurs [42]. A retrospective cohort study of prognostic factors, which analyzed all cases of adults hospitalized for severe COVID-19 in the state of Rio Grande do Sul, Brazil, have shown that obesity largely impacted in-hospital case fatality rates among young adults and Black people contaminated by COVID-19 [43].

A recent study published, which aimed to analyze the survival of patients hospitalized with COVID-19 and its associated factors in Brazil, has concluded that non-work-related infection, age group above or equal to 60 years, presence of chronic cardiovascular disease, chronic kidney disease, chronic neurological disease, smoking, obesity, and neoplasms were associated with a higher risk of death, and, therefore, a lower survival in Brazilian patients hospitalized with COVID-19 [9].

As for the criterion for diagnostic confirmation, laboratory confirmation was predominant. It is important to mention that the Central Public Health Laboratory of Espírito Santo (LACEN-ES) was enabled in March 2020 to perform COVID-19 diagnostic tests. Thus, there was the possibility for the state to offer expanded tests in the territory of Espírito Santo, as well as to speed up the results and ensure the quality of the information produced. LACEN-ES received samples from all hospital profiles until private services were qualified and purchased by private hospitals [44].

The variable origin was significant in relation to the outcome under study. Among the deaths in public hospitals, a greater chance of death was observed when the origin is the Central and North Health Region. This region had two state hospitals as a reference for COVID-19 treatment. However, this health region has 29 municipalities, and is characterized by a population with lower socioeconomic status and greater dependence on the Unified Health System. An important factor to be highlighted is that this health region is the one with the lowest number of hospital units and low coverage by the Mobile Emergency Care Service (Serviço de Atendimento Móvel de Urgência—SAMU), which hinders both pre-hospital care and decreases the speed of access to available hospital beds [45].

## 5. Conclusions

The deaths that occurred in public hospitals were associated with socio-clinical characteristics. The analysis of deaths by COVID-19 is extremely relevant to support management in the planning of the Health Care Network.

It can be concluded in the present study that the time interval between diagnosis and an outcome of death can be impacted by several factors, such as: plasticity of the health system, better clinical management of patients and initiation of vaccination at the end of January 2021, which covered the age group with the highest incidence of deaths.

There was an association between deaths in public hospitals and socioeconomic characteristics.

It can be observed that during the pandemic the resilience of the Unified Health System (SUS) was tested in several ways and that the articulation of different actors was necessary for a positive intervention in face of this difficult scenario [44,46]. In the state of Espírito Santo the strategy of building field hospitals was not adopted. Thus, it was decided to expand the number of beds in hospitals of the state network and to contract providers of the philanthropic and private network only when necessary.

It should be highlighted that the public response to the pandemic required health system managers to adopt different strategies to expand the installed capacity of care [47], making the purchase of beds in the private service a reality not only in ES, but worldwide, within which there was the allocation of beds exclusively for COVID-19 patients [48,49].

The limitation of the study is the impossibility of identifying, from secondary data, the SUS patients who occupied beds in private and philanthropic hospitals. Although the information was produced from secondary data, the quality of the information is guaranteed by the investigation of deaths carried out by the team from the Situation Room for the fight against the COVID-19 pandemic of the State Department of Health of Espírito Santo. It is noteworthy that the data were reported by the hospitals where the death occurred, with validation of the information from the Central Laboratory of Espírito Santo and the Death Information System (Sistema de Informação de Óbito—SIM) [44].

Finally, it is suggested that further studies be carried out to deepen the discussion of the theme presented, addressing other types of hospitals (private and philanthropic).

## Figures and Tables

**Table 1 ijerph-19-14077-t001:** Distribution of the waves of incidence of deaths according to socio-clinical variables for patients admitted to public hospital. April 2020 to August 2021, Espírito Santo, Brazil.

		Public						*p* * Value
		First		Second		Third		
		n	%	n	%	N	%	
Age range	Young	11	0.63	3	0.27	8	0.31	<0.001
	Adult	447	25.67	259	23.17	919	35.66	
	Elderly	1283	73.69	856	76.57	1650	64.03	
Sex	Female	749	43.02	517	46.24	1240	48.12	0.085
	Male	992	56.98	601	53.76	1337	51.88	
Race/color	White	523	33.42	363	36.56	709	32.40	<0.001
	Black	158	10.10	91	9.16	205	9.37	
	Yellow	218	13.93	116	11.68	285	13.03	
	Brown (*Parda*)	663	42.36	422	42.50	988	45.16	
	Indian	3	0.19	1	0.10	1	0.05	
Education	Illiterate	143	15.26	71	11.58	92	6.57	<0.001
	Elementary School	387	41.30	311	50.73	622	44.40	
	High School	364	38.85	193	31.48	601	42.90	
	High Education	43	4.59	38	6.20	86	6.14	
Fever	Yes	1012	58.67	537	48.16	1294	50.25	<0.001
	No	713	41.33	578	51.84	1281	49.75	
Difficulty breathing	Yes	1123	64.65	678	60.75	1375	53.38	<0.001
	No	614	35.35	438	39.25	1201	46.62	
Flapping wing nose	Yes	58	3.37	17	1.53	38	1.47	0.774
	No	1662	96.63	1096	98.47	2539	98.53	
Cyanosis	Yes	62	3.60	60	5.38	55	2.14	0.042
	No	1659	96.40	1056	94.62	2521	97.86	
O_2_ saturation < 95%	Yes	941	54.27	572	51.21	1172	45.48	<0.001
	No	793	45.73	545	48.79	1405	54.52	
Coma	Yes	33	1.92	27	2.42	24	0.93	0.226
	No	1689	98.08	1089	97.58	2552	99.07	
Coughing	Yes	1097	63.59	618	55.43	1638	63.56	0.029
	No	628	36.41	497	44.57	939	36.44	
Sputum production	Yes	118	6.84	60	5.38	113	4.38	0.004
	No	1608	93.16	1055	94.62	2464	95.62	
Nasal or conjunctival congestion	Yes	86	5.00	65	5.83	179	6.95	<0.001
	No	1633	95.00	1049	94.17	2397	93.05	
Coryza	Yes	310	18.02	164	14.70	551	21.39	0.003
	No	1410	81.98	952	85.30	2025	78.61	
Sore throat	Yes	218	12.70	133	11.95	417	16.19	0.135
	No	1499	87.30	980	88.05	2158	83.81	
Nausea/Vomiting	Yes	135	7.84	90	8.07	262	10.17	0.434
	No	1587	92.16	1025	91.93	2314	89.83	
Cephalea	Yes	411	23.97	268	24.04	862	33.46	<0.001
	No	1304	76.03	847	75.96	1714	66.54	
Irritability/Confusion	Yes	49	2.85	27	2.42	29	1.13	0.010
	No	1672	97.15	1088	97.58	2547	98.87	
Adynamia/Weakness	Yes	567	33.02	368	33.03	863	33.49	0.051
	No	1150	66.98	746	66.97	1714	66.51	
ICU inpatient	Yes	945	60.58	710	78.98	1586	75.60	0.159
	No	615	39.42	189	21.02	512	24.40	
Chronic Pulmonary Disease	Yes	164	9.54	81	7.28	127	4.94	0.312
	No	1555	90.46	1032	92.72	2443	95.06	
Chronic Cardiovascular Disease	Yes	994	57.72	643	57.62	1134	44.12	<0.001
	No	728	42.28	473	42.38	1436	55.88	
Neoplasia (solid or hematological tumor)	Yes	30	1.75	17	1.53	19	0.74	<0.001
	No	1687	98.25	1095	98.47	2553	99.26	
Smoking	Yes	130	7.57	76	6.83	98	3.81	0.002
	No	1587	92.43	1036	93.17	2474	96.19	
Bariatric Surgery	Yes	3	0.17	0	0.00	0	0.00	0.004
	No	1715	99.83	1114	100.00	2572	100.00	
Obesity	Yes	156	9.08	95	8.52	251	9.76	<0.001
	No	1563	90.92	1020	91.48	2320	90.24	
Neoplasms	Yes	40	2.33	31	2.78	28	1.09	<0.001
	No	1678	97.67	1083	97.22	2542	98.91	
Chronic Neurological Disease	Yes	61	3.55	71	6.37	96	3.73	0.026
	No	1658	96.45	1043	93.63	2475	96.27	
Confirmation criterion	Laboratorial	1726	99.14	1078	96.42	2465	95.65	0.032
	Clinical Epidemiological	10	0.57	5	0.45	8	0.31	
	Clinical	4	0.23	4	0.36	14	0.54	
	Clinical-image	1	0.06	31	2.77	90	3.49	
Origin	Central/North	421	24.18	358	32.02	785	30.46	<0.001
	Metropolitan	1181	67.83	604	54.03	1469	57.00	
	South	123	7.06	122	10.91	260	10.09	
	Other States	16	0.92	34	3.04	63	2.44	
		Med	M (CI95%) (±SD) (IIQ)	Med	M (CI95%) (±SD) (IIQ)	Med	M (CI95%) (±SD) (IIQ)	<0.001 **
Time between diagnosis and death		16.0a	18.5 (17.9–19.1) (±13.2) (10–24)	17.0a	20.5 (19.5–21.5) (±17.8) (11–26)	20.0b	21.4 (20.9–21.9) (±12.7) (13–27)	

(*) Pearson’s or Fisher’s Exact chi-square test; (**) Friedman’s test; significant if *p* ≤ 0.050; Med—Median; M—Mean; SD—Standard deviation.

**Table 2 ijerph-19-14077-t002:** Association of the public bond with the socio-clinical variables for the first wave of death incidence. From April 2020 to August 2021, Espírito Santo, Brazil.

Variable Dependent—Relationship/Bond (Public)			*p* * Value	OR	95% CI for OR		Omnibus Test—χ^2^ (*p* Value)	Hosmer–Lemshow Test—χ^2^ (*p* Value)	Pseudo-R^2^
					Lower Limit	Upper Limit			
1st wave	Imunnodeficiency	No	-	1	-	-	166.7 (<0.001)	6.8 (0.600)	17.9%
		Yes	0.038	1.76	1.032	3.003			
	Obesity	Yes	-	1	-	-			
		No	0.046	1.26	1.004	1.592			
	Neoplasms	Yes	-	1	-	-			
		No	<0.001	4.06	2.808	5.855			

(*) Multiple logistic regression with forward selection method; OR—Odds Ratio; (1) reference category; significant if *p* ≤ 0.050. The reference category of the dependent variable is the other hospitals (philanthropic + private).

**Table 3 ijerph-19-14077-t003:** Association of the public bond with the socio-clinical variables for the second wave of death incidence. From April 2020 to August 2021, Espírito Santo, Brazil.

Variable Dependent—Bond (Public)			*p* * Value	OR	95% CI for OR		Omnibus Test—χ^2^ (*p* Value)	Hosmer–Lemshow Test—χ^2^ (*p* Value)	*Pseudo*-R^2^
					Lower Limit	Upper Limit			
	Origin	Central/North	<0.001	5.41	4.190	6.996			
		Metropolitan	<0.001	3.65	2.931	4.543			
		Other states	<0.001	3.74	1.796	7.793			
		South	-	1	-	-			
2nd wave	Education	Illiterate	<0.001	3.53	2.072	6.026	102.5 (<0.001)	7.9 (0.435)	10.2%
		Elementary School	<0.001	3.15	2.065	4.801			
		High School	<0.001	2.11	1.374	3.239			
		High Education	-	1	-	-			
	O_2_ Saturation < 95%	No	-	1	-	-			
		Yes	<0.001	1.54	1.221	1.943			
	Chronic Neurological Disease	Yes	-	1	-	-			
		No	0.023	1.68	1.074	2.625			

(*) Multiple logistic regression with forward selection method; OR—Odds Ratio; (1) reference category; significant if *p* ≤ 0.050. The reference category of the dependent variable is the other hospitals (philanthropic + private).

**Table 4 ijerph-19-14077-t004:** Association of the public bond with the socio-clinical variables for the third wave of death incidence. From April 2020 to August 2021, Espírito Santo, Brazil.

Variable Dependent—Bond (Public)			*p* * Value	OR	95% CI for OR		Omnibus Test—χ^2^ (*p* Value)	Hosmer–Lemshow Test—χ^2^ (*p* Value)	*Pseudo*-R^2^
					Lower Limit	Upper Limit			
3rd wave	Origin	Central/North	<0.001	3.90	2.662	5.717			
		Metropolitan	<0.001	2.55	1.828	3.568			
		Other States	0.147	1.94	0.793	4.740			
		South	-	1	-	-			
	Race/color	White	-	1	-	-	258.3 (<0.001)	6.9 (0.544)	12.4%
		Black	<0.001	1.97	1.450	2.665			
		Yellow	<0.001	2.19	1.584	3.026			
		Brown (*Parda*)	<0.001	1.38	1.154	1.644			
		Indian	0.667	1.84	0.113	29.948			
	Education	Illiterate	<0.001	2.34	1.525	3.575			
		Elementary School	<0.001	2.60	1.932	3.502			
		High School	<0.001	2.26	1.679	3.028			
		High Education	-	1	-	-			
	Difficulty breathing	No	-	1	-	-			
		Yes	<0.001	1.49	1.263	1.753			
	Nasal or conjunctival congestion	Yes	-	1	-	-			
		No	0.003	1.50	1.143	1.957			
	Irritability/Confusion	Yes	-	1	-	-			
		No	0.004	2.58	1.360	4.911			
	Adynamia/Weakness	Yes	-	1	-	-			
		No	0.016	1.23	1.039	1.454			
	Chronic Cardiovascular Disease	Yes	-	1	-	-			
		No	0.001	1.34	1.125	1.601			
	Diabetes Mellitus	No	-	1	-	-			
		Yes	0.025	1.25	1.028	1.515			
	Neoplasms	Yes	-	1	-	-			
		No	0.001	2.97	1.572	5.612			
	Origin	Central/North	<0.001	3.56	2.666	4.761			
		Metropolitan	<0.001	1.89	1.484	2.410			
		Other States	0.105	1.77	0.888	3.524			
		South	-	1	-	-			

(*) Multiple logistic regression with forward selection method; OR—Odds Ratio; (1) reference category; significant if *p* ≤ 0.050. The reference category of the dependent variable is the other hospitals (philanthropic + private).

## Data Availability

Not applicable.

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
