# Peer review of "COVID-19 Mortality in Public Hospitals in a Brazilian State: An Analysis of the Three Waves of the Pandemic"

_ijerph, 2022, doi:10.3390/ijerph192114077_

Round 1

Reviewer 1 Report

Dell’Antonio et al in their manuscript entitled “COVID-19 mortality in public hospitals in a Brazilian state: an analysis of the three waves of the pandemic” provide a retrospective analysis of Brazilian COVID-19 deaths in public hospitals in a Espirito Santo state, stratified by April 1, 2020, and September 4, 2020 (first 95 wave), September 5, 2020, and February 13, 2021 (second wave), and February 14, 2021, 96 and August 31, 2021 (third wave) to test their association with socio-clinical variables. The study was also approved by the Research Ethics Committee. The paper provides important and statistically valid information regarding the correlation between COVID-19 deaths and socio-clinical variables.

There are some points to be adressed or answered:

1. I strongly suggest that the authors expand the references in the introduction with other published papers, worldwide, regarding the correlation between death from COVID-19 and clinical symptoms, pre-existing diseases and socioeconomic conditions.

2. Please, better explain the choices of tests based on the data collected. The reader of this article may benefit from having, for instance, time series and normality test plots in a supplementary material published with this article. If possible, also make all the raw data available.

3. Table 2 should be divided into 3 tables, for each of the waves, to make it more understandable.

4. The authors says: In In order to evaluate the association of the public bond with socio-clinical variables for the three waves of death incidence, a comparison was made between the data from public hospitals and the grouped data from the other hospital bonds (private and philanthropic) (line 148). But after in the manuscript they say: The limitation of the study is the impossibility of identifying, from secondary data, the SUS patients who occupied beds in private and philanthropic hospitals (line 339). It is not clear, for me, the confidence level of the results of private and philantropic hospitals after the second statement. Can you explain it better?

5. Lines 335 to 348. This text should be in Conclusion.

6. Please add also some results highlights to improve the Conclusion, as the information shown in the abstract.

Author Response

Comments and Suggestions for Authors

Dell’Antonio et al in their manuscript entitled “COVID-19 mortality in public hospitals in a Brazilian state: an analysis of the three waves of the pandemic” provide a retrospective analysis of Brazilian COVID-19 deaths in public hospitals in a Espirito Santo state, stratified by April 1, 2020, and September 4, 2020 (first 95 wave), September 5, 2020, and February 13, 2021 (second wave), and February 14, 2021, 96 and August 31, 2021 (third wave) to test their association with socio-clinical variables. The study was also approved by the Research Ethics Committee. The paper provides important and statistically valid information regarding the correlation between COVID-19 deaths and socio-clinical variables.

Response: Ok. Thank you very much for your careful review as well as your positive feedback . We really appreciate it!

There are some points to be addressed or answered:

  1. I strongly suggest that the authors expand the references in the introduction with other published papers, worldwide, regarding the correlation between death from COVID-19 and clinical symptoms, pre-existing diseases and socioeconomic conditions.

Response: Ok. Done. Two new references were added in the introduction section.

  1. Please, better explain the choices of tests based on the data collected. The reader of this article may benefit from having, for instance, time series and normality test plots in a supplementary material published with this article. If possible, also make all the raw data available.

Response: Ok. Thank you for this valuable suggestion. We have added these information in detail on methods section as per recommended.

It is possible to obtain all updated public covid data by accessing the “Covid-19 State of Espírito Santo Panel” developed to provide transparency to the data, available on the website: https://coronavirus.es.gov.br/painel-covid- 19-es.

As for the raw data of the study, as they have sensitive patient information, it is impossible to share them.

  1. Table 2 should be divided into 3 tables, for each of the waves, to make it more understandable.

Response: Ok. We agreed with you. Table 2 was divided into 3 tables. Thanks!

  1. The authors says: In In order to evaluate the association of the public bond with socio-clinical variables for the three waves of death incidence, a comparison was made between the data from public hospitals and the grouped data from the other hospital bonds (private and philanthropic) (line 148). But after in the manuscript they say: The limitation of the study is the impossibility of identifying, from secondary data, the SUS patients who occupied beds in private and philanthropic hospitals (line 339). It is not clear, for me, the confidence level of the results of private and philantropic hospitals after the second statement. Can you explain it better?

Response: Yes. Considering that we have all the data on deaths that occurred in hospitals in the state of Espírito Santo and we extract them from the three waves of the pandemic.

  1. Lines 335 to 348. This text should be in Conclusion.

Response: Ok. Done.

  1. Please add also some results highlights to improve the Conclusion, as the information shown in the abstract.

Response: Ok. Done.

Author Response

Response:  First of all, I would like to thank you for your very detailed and thoughtful review. Before going directly to the highlighted points, I would like to clarify that the authors are professionals who work at the State Health Department of Espírito Santo, specifically in the Covid-19 Situation Room of the Health Surveillance, thus having access to the raw data and performing the qualification of the data.

Regarding the data on deaths by Covid-19, there was a team responsible for the detailed investigation of the same with the municipalities and health establishments where the deaths occurred. At the time of hospitalization, suspected patients were tested and notified for Covid-19, and therefore, previous information from patients and clinics at the time of hospitalization. It should be highlighted that for deaths that occurred outside the health services, the investigation was carried out and the compulsory notification form was filled in, thus making it possible to obtain the individual's previous socio-clinical information.

In this study, only deaths that occurred in hospitals were analyzed, excluding deaths that occurred in other places (home, emergency care, ambulances, other states, etc). In this sense, we made the correction suggested in the methodology where there was mention of previous hospitalization or not.

Regarding the statistical programs used, the data were grouped and organized in STATA and analyzed in SPSS, now being described more clearly in the methodology. New information about the statistical tests used was also included in the methodology, as well as a reference was added. Friedman's test was used due to non-normal distribution. Knowing that the mean is a volatile measure. There are outliers, just as the linear regression errors did not assume some of its assumptions, quantile regression can better estimate the association. Linear regression was performed and transformations were also performed, but in the end it was not possible to find a normal distribution of errors.

Regarding the stratification of hospitals in terms of public, private or philanthropic bonds, the situation room team standardized the profile of each hospital according to its legal nature and classified them with information only on cases that resulted in death. Thus, for this study, we compared the deaths that occurred in public versus non-public hospitals (this group is composed of patients from private and philanthropic hospitals).

Considering that we received daily clinical and epidemiological information from hospitalized patients and even though our main instrument for collecting information was the Covid-19 notification form and that it only contains information on the place of hospitalization and place of death, it is not it is possible to differentiate between patients who is a SUS patient or not.That is, if a SUS patient was hospitalized in a private hospital due to the purchase of beds made by the state management, it is not possible to identify this patient. Thus, we point out this fact as a limitation of the study. It is noteworthy that no study will be able to make this distinction because there is no instrument that assesses this variable.

Regarding the questioning about the cause of death, it is necessary to emphasize that in the Covid-19 notification form there are 4 possible outcomes: 1- cure, 2- death from Covid-19, 3- death from other causes and 9- ignored . Thus, all patients who died underwent a process of epidemiological investigation and diagnostic confirmation associated with this outcome, being classified as deaths from Covid or death from other causes (in this study, only patients who died from Covid were analyzed).

In this way, our team qualified all deaths, confirming those with laboratory and cynical criteria and discarding those that did not meet clinical, epidemiological or laboratory criteria. I emphasize that just the fact that Covid was included in the Death Certificate as the underlying cause was not enough to be classified as such by the epidemiological surveillance.

Regarding the analysis of the socio-clinical variables, I emphasize that, based on the process and epidemiological investigation described briefly above, the composition of the group of patients classified with the outcome deaths by Covid was obtained. Thus, the analysis of socio-clinical variables was carried out only for this group of patients.

Regarding the notes made in relation to Table 1, it presents the description of the profile of deaths that occurred in public hospitals, stratified by waves of occurrence, being described in the text in the first paragraphs of the results section.

As you suggested, additional measures were inserted in Table 1 (mean, median, standard deviation, confidence interval and quantile interval of time between diagnosis and death).

In tables 2, 3 and 4 (Table 2 divided into 3 for better understanding as requested by reviewer 1) the variables that showed statistical significance in the first analysis (Table 1) were subjected to statistical analysis to assess the correlation between them and the waves of pre-established occurrence.

Thank you so much!

Reviewer 3 Report

The authors have carried out the analysis and interpreted the results well.

The statistical techniques and tests used in the present paper needs to be elaborated a little bit.

The Multiple multinomial regression with the forward variable selection method has been used in the present analysis. As  so many variables have been considered in the analysis, it is likely that multicollineariy will arise.

How did the authors take care of this phenomena?

The authors have mentioned about many tests but their references are missing.

The authors have reported that SPSS and STAT have been used for analysis. It is better to specify which one is used for which analysis for better understanding of the readers.

Author Response

Comments and Suggestions for Authors

The authors have carried out the analysis and interpreted the results well.

Response: Thank you for your revision.

The statistical techniques and tests used in the present paper needs to be elaborated a little bit.

Response: Ok. Thank you for this valuable suggestion. We have added these information in detail on methods section as per recommended.

The Multiple multinomial regression with the forward variable selection method has been used in the present analysis. As  so many variables have been considered in the analysis, it is likely that multicollineariy will arise.How did the authors take care of this phenomena?

 Response: When performing the statistical analysis, care was taken to ensure that there were no problems with missing variables, the form of the model was verified and whether it is correctly specified, and the presence of outliers was verified to ensure that the variables were measured correctly.

The authors have mentioned about many tests but their references are missing.

Response: Ok. We have added the references as per suggested. Thanks!

The authors have reported that SPSS and STATA have been used for analysis. It is better to specify which one is used for which analysis for better understanding of the readers.

Response: Ok. The adjustments were made.

All co-authors have agreed to the resubmission with these revisions. The manuscript is consistent with the Guidelines for Authors of the IJERPH.

We are looking forward to hearing from you soon, and we hope to have our manuscript accepted for publication in this renowned journal.

Yours Sincerely,

The authors

Round 2

Reviewer 1 Report

Dear authors. Thank you for the revised manuscript. All the comments were properly addressed.